# BRIDGING THE GAP BETWEEN SL AND TD LEARNING VIA Q-CONDITIONED MAXIMIZATION

## ABSTRACT

Recent research highlights the efficacy of supervised learning (SL) as a methodology within reinforcement learning (RL), yielding commendable results. Nonetheless, investigations reveal that SL-based methods lack the stitching capability typically associated with RL approaches such as TD learning, which facilitate the resolution of tasks by stitching diverse trajectory segments. This prompts the question: *How can SL methods be endowed with stitching property and bridge the gap with TD learning?* This paper addresses this challenge by exploring the maximization of the objective in the goal-conditioned RL. We introduce the concept of Q-conditioned maximization supervised learning, grounded in the assertion that the goal-conditioned RL objective is equivalent to maximizing the expected Q-function under given goal distribution, thus embedding Q-function maximization into traditional SL-based methodologies. Building upon this premise, we propose **G**oal-**C**onditioned ***Rein***forced **S**upervised **L**earning (**GC*Rein*SL**), which enhances SL-based approaches by incorporating maximizing Q-function. **GC*Rein*SL** emphasizes the maximization of the Q-function during the training phase to predict the maximum Q-function within the distribution. This optimized in-distribution Q-function is then employed during the inference phase to guide the selection of optimal actions. We demonstrate that **GC*Rein*SL** enables SL methods to exhibit stitching property, effectively equivalent to applying goal data augmentation to SL methods. Experimental results on offline datasets designed to evaluate stitching capability show that our approach not only effectively selects appropriate goals across diverse trajectories but also outperforms previous works that applied goal data augmentation to SL methods.

## 1 INTRODUCTION

Recently, numerous methods that frame reinforcement learning RL as a purely SL problem (Schmidhuber, 2020; Chen et al., 2021; Emmons et al., 2021; Chane-Sane et al., 2021a) function by correlating input states and desired goals with optimal actions. These techniques assign labels to state-action pairs based on future outcomes (e.g., achieving a goal) derived from offline datasets, subsequently maximizing the likelihood of these actions as optimal for producing the intended results. Collectively termed outcome-conditioned behavioral cloning algorithms (OCBC), these approaches have exhibited commendable performance on standard offline benchmarks (Emmons et al., 2021). Nevertheless, recent investigations (Yang et al., 2023; Ghugare et al., 2024) have highlighted a critical shortcoming of these SL methodologies: the lack of trajectory stitching capability. This property, commonly found in temporal-difference (TD)-based RL algorithms employing dynamic programming (e.g., CQL(Kumar et al., 2020), and IQL(Kostrikov et al., 2021a)), is vital for addressing tasks that require the integration of multiple trajectory segments. Thus, enhancing OCBC methods to incorporate this characteristic and bridging the gap with TD approaches has emerged as a significant area of research.

In this paper, we examine this issue within goal-conditioned RL, focusing on navigating between certain state-goal pairs that, while not co-occurring during training, are present in isolation. In sparse-reward goal-conditioned RL, TD-based RL methods often face challenges such as instability during training due to difficulties in accurately estimating the value function, inefficiencies in optimization (Van Hasselt et al., 2018; Kumar et al., 2019a), and high sensitivity to hyperparameters (Henderson et al., 2018). In contrast, OCBC methods are simpler, more efficient, and free from these issues,

making the development of novel OCBC approaches highly valuable. However, OCBC lacks the critical trajectory stitching property inherent to TD-based RL methods. Addressing this limitation to enable stitching and bridge performance gaps in challenging environments is a key focus of current research. We have observed that certain sequence modeling (Yamagata et al., 2023a; Wu et al., 2023; Zhuang et al., 2024) techniques are enabling Decision Transformer (DT) (Chen et al., 2021) within OCBC methods to acquire stitching property. However, these methods are primarily effective within goal-conditioned scenarios. Drawing motivation and inspiration from state-of-the-art max-return sequence modeling method (Zhuang et al., 2024), we propose the concept of Q-conditioned maximization supervised learning within the context of goal-conditioned RL. Specifically, since the objective in goal-conditioned RL is equivalent to maximizing the expected Q-function across all possible goals under the given goal distribution, we commence in Section 4.1 by examining a maze example to illustrate the detrimental impact of naively setting the Q-function to highest possible value on trajectory stitching. An illustrative example, shown in Fig. 1, highlights the relationship between a failing trajectory (with $Q = 0$, where the agent starts from the initial state but fails to reach the final goal) and a successful trajectory (with $Q = 1$, where the agent reaches the final goal but does not originate from the initial state). Ideally, the Q-function should start at $0$ and shift to $1$ when transitioning to the successful trajectory. This requirement contrasts with the oversimplified approach of artificially assigning a Q-function of $1$.

And then we propose the concept of Q-conditioned maximization supervised learning, a framework that embeds the maximization of Q-function into supervised learning. This approach aims not only to maximize the probability of selecting appropriate actions but also to predict the highest attainable in-distribution Q-function. To achieve this, we utilize expectile regression (Aigner et al., 1976; Sobotka & Kneib, 2012), which seeks to ensure that the predicted Q-function closely approximates the maximum Q-function that can be realized from the available historical trajectory. In the inference pipeline, the model first predicts the current maximum Q-function and then identifies the best action based on the offline dataset distribution, guided by this predicted maximum. Our findings indicate that Q-conditioned maximization supervised learning acts as a form of goal data augmentation for OCBC methods, leading to substantial improvements in their stitching capability. Additionally, we present **G**oal-**C**onditioned **Rein**forced **S**upervised **L**earning (**GC*Rein*SL**), which implements Q-conditioned maximization supervised learning for OCBC methods, including DT (Chen et al., 2021) and Reinforcement Learning via Supervised Learning (RvS) (Emmons et al., 2021). This framework reinforces supervised learning through the maximization of the Q-function. In scenarios involving trajectory stitching, as demonstrated in Fig. 1, GCReinSL typically predicts a value of 0 at the starting point and transitions to a prediction of 1 upon switching to a successful trajectory, reflecting the predicted in-distribution maximum Q-function.

We briefly summarize our main contributions as follows: (1) Inspired by max-return sequence modeling (Zhuang et al., 2024), we propose a novel supervised learning framework in goal-conditioned RL based on our concept of Q-conditioned maximization, which endows OCBC methods with stitching ability. (2) We demonstrate that **GC*Rein*SL** is equivalent to goal data augmentation for OCBC methods. (3) Experimental results in Ghugare et al. (2024) offline datasets, designed to test stitching ability, show that **GC*Rein*SL** not only significantly enhances the stitching capability of OCBC methods but also outperforms relevant goal data augmentation works. Additionally, in the goal-conditioned D4RL (Fu et al., 2020) offline datasets, our method continues to outperform related sequence modeling methods which also perform trajectory stitching.

## 2 RELATED WORK

**Goal-conditioned RL** This paper focus on goal-conditioned RL, a topic explored extensively in prior research through various methodologies. Approaches such as conditional supervised learning (Ding et al., 2019; Gupta et al., 2020; Lynch et al., 2020; Ghosh et al., 2021; Emmons et al., 2021), actor-critic frameworks (Andrychowicz et al., 2017; Nachum et al., 2018; Zhu et al., 2021; Chane-Sane et al., 2021b), model-based strategies (Schmeckpeper et al., 2020; Charlesworth & Montana, 2020; Mendonca et al., 2021), and distance metric learning (Tian et al., 2020; Nair et al., 2020; Durugkar et al., 2021; Liu et al., 2023a; Wang et al., 2023; Reichlin et al., 2024) have been employed to learn goal-conditioned policies. These methods have demonstrated success across diverse tasks, including real-world robotic systems (Ma et al., 2022; Shah et al., 2022; Zheng et al., 2023a). Unlike techniques that depend on manually defined reward or distance functions, our approach builds on a

self-supervised formulation of goal-conditioned RL, treating the task as one of predicting future state visitation (Eysenbach et al., 2020; 2022b; Zheng et al., 2023b; Ghugare et al., 2024).

**The Stitching Property** The concept of stitching, as discussed by Ziebart et al. (2008), is a characteristic property of TD-learning algorithms such as those described by Kumar et al. (2020); Kostrikov et al. (2021a), which employ dynamic programming techniques. This property enables these algorithms to integrate data from diverse trajectories, thereby improving their ability to handle complex tasks by effectively utilizing available data (Cheikhi & Russo, 2023). On the other hand, most SL-based RL methods lack this property. Brandfonbrener et al. (2022); Yang et al. (2023) provide examples where SL algorithms do not perform stitching and Ghugare et al. (2024) also indicates this from the perspective of combinatorial generalisation. In contrast, we use a simple maze example to illustrate this viewpoint from the perspective of maximizing the RL objective.

**Data Augmentation in RL** Data augmentation, as an efficient method for improving generalization ability, has been applied in RL (Lu et al., 2020; Stone et al., 2021; Kalashnikov et al., 2021; Hansen & Wang, 2021; Kostrikov et al., 2021b; Yarats et al., 2021) and SL (Shorten & Khoshgoftaar, 2019). We have noticed that some methods (Char et al., 2022; Yamagata et al., 2023b; Paster et al., 2023) use dynamic programming to enhance existing trajectories to improve the performance of SL algorithms. However, they still require dynamic programming. Another methods which are very similar to ours is to only perform data augmentation for SL (Yang et al., 2023; Ghugare et al., 2024). However, they may have the problem of not being able to correctly provide the augmented goal data such as unreachable goals. Unlike these two methods, we approach from the perspective of maximizing the goal-conditioned RL objective and endow the SL method with the ability to stitch trajectories, providing agents with a more reasonable selection of augmented goals.

## 3 PRELIMINARIES

### 3.1 GOAL-CONDITIONED RL IN CONTROLLED MARKOV PROCESS

We will study the problem of goal-conditioned RL in a controlled Markov process with states $s \in \mathcal{S}$, actions $a \in \mathcal{A}$. The dynamics are $p(s' \mid s, a)$, the initial state distribution is $p_0(s_0)$, the discount factor is $\gamma$, and a reward function $r(s, a, g)$ for each goal. The goal-conditioned policy $\pi(a, \mid s, g)$ is conditioned on a pair of state and goal $s, g \in \mathcal{S}$.

We denote the $t$-step action-conditioned policy distribution $p_t^\pi(s_t \mid s_0, a_0)$ as the distribution of states $t$ steps in the future given the initial state $s_0$ and action $a_0$ under $\pi$. For a policy $\pi$, define as the distribution over states visited after exactly $t$ steps. We define the discounted state occupancy distribution as:

$$p_+^\pi(s_{t+} \mid s, a) \triangleq (1 - \gamma) \sum_{t=0}^\infty \gamma^t p_t^\pi(s_{t+} \mid s, a), \tag{1}$$

where $s_{t+}$ is the variable that specifies a future state corresponding to the discounted state occupancy distribution. For a given distribution over goals $g \sim p_\mathcal{G}$, the objective of the policy $\pi$ is to maximize the probability of reaching the goal $g$ in the future:

$$\max_{\pi(\cdot \mid \cdot, \cdot)} \mathbb{E}_{p_0(s_0) p_\mathcal{G}(g) \pi(a_0 \mid s_0, g)} \left[ p_+^\pi(g \mid s_0, a_0) \right]. \tag{2}$$

Following prior work (Eysenbach et al., 2020; Chane-Sane et al., 2021b; Blier et al., 2021; Rudner et al., 2021; Eysenbach et al., 2022b; Bortkiewicz et al., 2024), we define the reward function $r(s, a, g)$ for each goal as the probability of reaching the goal at the next time step:

$$r(s_t, a_t, g) \triangleq (1 - \gamma) p(s_{t+1} = g \mid s_t, a_t). \tag{3}$$

And the Q-function can be defined for a policy $\pi(\cdot \mid \cdot, g)$:

$$Q^\pi(s, a, g) \triangleq \mathbb{E}_{\pi(\cdot \mid g)} \left[ \sum_{t=0}^\infty \gamma^t r(s_t, a_t, g) \mid {}^{s_0 = s,}_{a_0 = a} \right]. \tag{4}$$

**Theorem 3.1** (Rephrased from Proposition 1 of Eysenbach et al. (2022b)). *The Q-function for the goal-conditioned reward function in Eq. (4) is equivalent to the probability of goal $g$ under the discounted state occupancy distribution:*

$$Q^\pi(s, a, g) = p_+^\pi(s_{t+} = g \mid s, a). \tag{5}$$

The proof is in Appendix A.1. This proposition indicates that Q-function is equivalent to the discounted state occupancy distribution. Thus, from Eq. (2) and Eq. (5), we can conclude that the objective of the policy $\pi$ in goal-conditioned RL is equivalent to maximizing the expected Q-function over all possible goals under the given goal distribution $p_{\mathcal{G}}(g)$.

**Remark 1.** *Translating rewards to probabilities simplifies the analysis of goal-conditioned RL problem and allows probabilistic estimation methods (e.g., VAE (Kingma & Welling, 2014)) to be repurposed for Q-function estimation.*

Our work focuses on the **offline** goal-conditioned RL setting (Levine et al., 2020), the agent can only access a static offline dataset $\mathcal{D}$ and cannot interact with the environment. The offline dataset $\mathcal{D}$ can be collected by some unknown policies (Levine et al., 2020; Prudencio et al., 2023). We can express the offline dataset as $\mathcal{D} := \{\tau_i\}_{i=1}^{N}$ (Ghugare et al., 2024), where $\tau_i := \left\{ <s_0^i, a_0^i, r_0^i>, <s_1^i, a_1^i, r_1^i>, ..., <s_T^i, a_T^i, r_T^i> \right\}$ is the goal-conditioned trajectory and $N$ is the number of stored trajectories. In each $\tau_i$ for $i \in 1, ..., N$, $s_0^i \sim p_0(s_0)$.

### 3.2 OUTCOME CONDITIONAL BEHAVIORAL CLONING (OCBC) METHODS

We present empirical results using a simple and popular class of goal-conditioned RL methods: Outcome conditional behavioral cloning (Eysenbach et al., 2022a) (DT (Chen et al., 2021), URL (Schmidhuber, 2020), RvS (Emmons et al., 2021), GCSL (Chane-Sane et al., 2021a) and many others (Sun et al., 2019; Kumar et al., 2019b)). These SL methods take as input the offline dataset $\mathcal{D}$ and learn a goal-conditioned policy $\pi(a \mid s, g)$ using a maximum likelihood objective:

$$\max_{\pi(\cdot|\cdot,\cdot)} \mathbb{E}_{(s,a,g)\sim\mathcal{D}} \left[ \log \pi(a \mid s, g) \right]. \tag{6}$$

## 4 METHODOLOGY

In this section, we start with a simple maze example to illustrate why classical OCBC methods and the naive Q-conditioned maximization approach are unlikely to solve the trajectory stitching problem. And then we employ a VAE as a neural probability estimation model to approximate the Q-function. Further, we introduce the concept of Q-conditioned maximization supervised learning and theoretically demonstrate that this paradigm can achieve maximum Q-function without encountering out-of-distribution (OOD) issues. We also demonstrate that Q-conditioned maximization supervised learning is equivalent to goal data augmentation for OCBC methods. Finally, we outline the implementation details of our Q-conditioned maximization supervised learning, **GC*Rein*SL**, focusing on three key aspects: the model architecture, the loss function utilized during training, and the inference pipeline.

### 4.1 TRAJECTORY STITCHING EXAMPLE

In the offline RL literature, trajectory stitching has garnered significant attention. Ideally, an offline agent should be able to combine overlapping suboptimal trajectories into optimal ones (Kostrikov et al., 2021a; Liu et al., 2023b). Both theoretical (Ghugare et al., 2024) and empirical studies (Yang et al., 2023) have demonstrated that SL methods lack the ability to perform effective stitching. The following example provides a detailed explanation of this limitation.

**Example** The Fig. 1 depicts a toy maze, where $s_0^1$ is the starting state, $g$ is the final goal with reward $r = 1$, $g'$ is a boom goal with $r = -1$ and other states are all $r = 0$. The offline dataset contains two trajectories one trajectory $\tau_1$ starts from the initial state $s_0$ and reach the goal $g_1$ but doesn't reach the final goal while another $\tau_2$ reaches the final goal $g$ but doesn't start from $s_0^1$. $s_t$ is the intersection of two trajectories and $g'$ is the boom goal that we aim to avoid reaching. Trajectory stitching expects the agent can follow the first half of $\tau_1$ (from starting state $s_0^1$ to $s_t$) and then take the second half of $\tau_2$ (from $s_t$ to the goal $g$) to reach the goal. We first explain why the typical OCBC methods might fail.

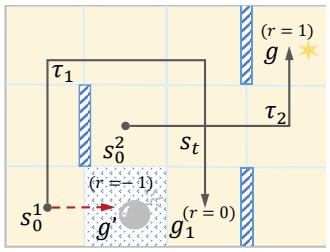

Figure 1: A maze example for trajectory stitching analysis.

If we set initial Q-function as $\hat{Q}_0 = 0$ at the starting state, the agent will smoothly reach the intersection state $s_t$. However, since Q-function is still zero $\hat{Q}_t = 0$ at the state $s_t$, OCBC methods will reach the state $g_1$ rather then $g$. Only when $\hat{Q}_t = 1$, OCBC methods is possible to follow $\tau_2$. But $\hat{Q}_t = 1$ is impossible to obtain given $\hat{Q}_0 = 0$. If we apply the naive max approach and set the initial $\hat{Q}_0 = 1$, the agent might directly walk towards the boom goal $g'$ $(r = -1)$ because $\hat{Q}_0 = 1$ is the OOD Q-function for the starting state.

If the OCBC methods are endowed with capability to maximize the Q-function like goal-conditioned RL, Let's see what might happen. At the starting state $s_0^1$, only $\tau_1$ is contained in dataset so the model will predict $\hat{Q}_0 = 0$. When offline agent comes to the intersection $s_t$, the latter segments of both trajectories are available. If the OCBC methods are able to maximize Q-function, then $\tau_2$ is more likely to be selected since the Q-function $Q = 1$ is larger. This inspires us to bring the capability of maximizing Q-function back into supervised learning.

### 4.2 Q-FUNCTION ESTIMATION WITH VAE

The central aim of goal-conditioned RL is to identify the best action for a given state and goal by maximizing the Q-function. To achieve this, the first task is to accurately estimate the Q-function. Drawing on previous research (Wu et al., 2022) and Theorem 3.1, we implement a Variational Autoencoder (VAE) architecture as a probabilistic modeling tool. More specifically, we apply a Conditional Variational Autoencoder (CVAE) (Sohn et al., 2015) for probability estimation. In our framework, the probability $p_+^\pi(g \mid s_0 = s, a)$ is modeled by a Deep Latent Variable Model, expressed as $p_\psi(g|s,a) = \int p_\psi(g|z,s,a)p(z|s,a)\mathrm{d}z$, with a prior distribution $p(z|s,a) = \mathcal{N}(\mathbf{0}, I)$. Although directly calculating the marginal likelihood $p_\psi(g|s,a)$ is computationally infeasible, VAE utilizes an approximate posterior $q_\varphi(z|s,a,g) \approx p_\psi(z|s,a,g)$, enabling joint optimization of $\psi$ and $\varphi$ parameters via the evidence lower bound (ELBO):

$$
\begin{aligned}
\log p_\psi(g|s,a) &\geq \mathbb{E}_{q_\varphi(z|s,a,g)} \left[ \log \frac{p_\psi(g, z|s, a)}{q_\varphi(z|s, a, g)} \right] \\
&= \mathbb{E}_{q_\varphi(z|s,a,g)} \left[ \log p_\psi(g|z, s, a) \right] - \mathrm{KL} \left[ q_\varphi(z|s, a, g) \| p(z|s, a) \right] \\
&\stackrel{\mathrm{def}}{=} -\mathcal{L}_{\mathrm{ELBO}}(s, a; \varphi, \psi).
\end{aligned}
\tag{7}
$$

After training this VAE, we can approximate the probability $p_+^\pi(g \mid s, a)$ in Eq. (5) by $-\mathcal{L}_{\mathrm{ELBO}}$. To obtain an estimation with lower bias between $\log p_\psi(g|s,a)$ and $p_+^\pi(g \mid s, a)$ in Eq. (5), we use the importance sampling technique following Rezende et al. (2014); Kingma & Welling (2019); Wu et al. (2022):

$$
\begin{aligned}
\log p_\psi(g|s,a) &= \log \mathbb{E}_{q_\varphi(z|s,a,g)} \left[ \frac{p_\psi(g, z|s, a)}{q_\varphi(z|s, a, g)} \right] \\
&\approx \mathbb{E}_{z^{(l)} \sim q_\varphi(z|s,a,g)} \left[ \log \frac{1}{L} \sum_{l=1}^{L} \frac{p_\psi(a, g, z^{(l)}|s)}{q_\varphi(z^{(l)}|s, a, g)} \right] \\
&\stackrel{\mathrm{def}}{=} \widehat{\log p_+^\pi}(g|s, a; \varphi, \psi, L).
\end{aligned}
\tag{8}
$$

From the reward and probability transformation in Theorem 3.1, the value of the Q-function can be derived.

### 4.3 Q-CONDITIONED MAXIMIZATION SUPERVISED LEARNING

After estimating the Q-function, we aim to equip supervised learning with additional maximizing Q-function objective , analogous to the methods employed in RL. And during inference, the supervised learning can select optimal action conditioned on the in-distribution maximized Q-function. We introduce the expectile regression as Q-function loss to achieve this.

Expectile regression (Newey & Powell, 1987) is well studied in applied statistics and econometrics and has been introduced into offline RL recently (Kostrikov et al., 2021a; Wu et al., 2023; Zhuang et al., 2024). Specifically, the Q-function loss based on the expectile regression is as follows:

$$
\mathcal{L}_Q^m = \mathbb{E}_{(s,a,g) \in \mathcal{D}} \left[ |m - \mathbb{1}(\Delta Q < 0)| \Delta Q^2 \right],
\tag{9}
$$

here $Q = Q^\pi(s, a, g)$, $\Delta Q = Q - \hat{Q}$ and $\hat{Q}$ can come from the supervised learning model (e.g, DT model can independently predict both the Q-function and the corresponding actions). Here $m \in (0, 1)$ is the hyperparameter of expectile regression. When $m = 0.5$, expectile regression degenerates into standard regression, also MSE loss. $\hat{Q}$, which aligns with the asymmetric curves in Fig. 2.

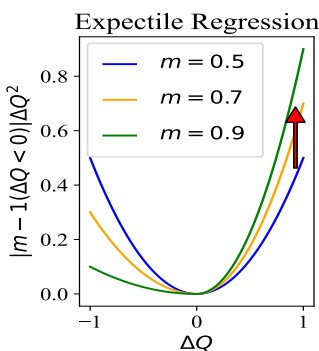

Figure 2: Illustration of weight.

But when $m > 0.5$, this asymmetric loss will give more weights to the $Q$ larger than $\hat{Q}$. Besides, The red arrow shows the weight increases as the $m$ becomes larger. In other words, the predicted Q-function $\hat{Q}$ will approach larger $Q$.

To unveil what the Q-function loss function has learned and offer a formal elucidation of its role, we introduce the following theorem:

**Theorem 4.1.** *Suppose Q-function is predict by the model itself, we first define* $\mathbf{SG} \doteq (s, g, a, Q)$*. For* $m \in (0, 1)$*, denote* $\mathbf{Q}^m(\mathbf{SG}) = \arg\min \mathcal{L}_Q^m(\mathbf{SG})$*, then we have*

$$\lim_{m \to 1} \mathbf{Q}^m(\mathbf{SG}) = Q_{max},$$

*where* $Q_{max} = \max_{\mathbf{a} \sim \mathcal{D}} Q(s, a, g)$ *denotes the maximum Q-function with actions from offline dataset.*

The proof is in Appendix A.2. In other words, Theorem 4.1 indicates the loss $\mathcal{L}_Q^m$ will make the model predict the maximum Q-function when $m \to 1$, which is similar to the maximizing objective in goal-conditioned RL.

**Corollary 1.** *The concept of Q-conditioned maximization supervised learning is equivalent to applying goal data augmentation for supervised learning (SL) methods, enabling it to exhibit stitching property.*

The proof is in Appendix A.3. Corollary 1 indicates that Q-conditioned maximization supervised learning can select state-goal pairs formed by trajectory stitching, which is consistent with the discussion presented in Section 4.1.

### 4.4 IMPLEMENTATION OF **GC*Rein*SL**

Now, we will focus on the specific implementation of **GC*Rein*SL**, describing the architecture input and output, training, and inference procedures. Specifically, this section describes the training and inference pipeline using two typical OCBC algorithms: DT and RvS. Other supervised learning algorithms can be implemented in a similar manner. The overall structure of **GC*Rein*SL** for DT is depicted in Fig. 3, with RvS being similar, differing only in terms of its architecture.

#### 4.4.1 **GC*Rein*SL** FOR DT

**Model Architecture** To accommodate the Q-conditioned maximization for DT (Chen et al., 2021), which predicts the maximum Q-function and utilizes it as a condition to guide the generation of optimal actions, we have positioned Q-function between state and goal. In detail, the input token sequence of **GC*Rein*SL** for DT and corresponding output tokens are summarized as follows:

$$\textbf{Input:} \quad \left\langle \cdots, sg_t^{(n)}, Q_t^{(n)}, a_t^{(n)} \right\rangle$$

$$\textbf{Output:} \quad \left\langle \hat{Q}_t^{(n)}, \hat{a}_t^{(n)}, \square \right\rangle$$

$sg_t^{(n)}$ represents a token formed by concatenating $s_t^{(n)}$ and $g_t^{(n)}$ (Schaul et al., 2015). When predicting the $\hat{Q}_t^{(n)}$, the model takes the current state $s_t^{(n)}$ and previous $K$ timesteps tokens $\langle sg, Q, a \rangle_{t-K}^{(n)} = \left( sg_{t-K+1}^{(n)}, Q_{t-K+1}^{(n)}, a_{t-K+1}^{(n)}, \cdots, sg_{t-1}^{(n)}, Q_{t-1}^{(n)}, a_{t-1}^{(n)} \right)$ into consideration. For the sake of simplicity, $\mathbf{SG}_{t-K}^{(n)}$ denotes the input $\left[ \langle sg, Q, a \rangle_{t-K}^{(n)}; sg_t^{(n)} \right]$. While the action prediction $\hat{a}_t$ is based on $\left( \mathbf{SG}_{t-K}^{(n)}, \mathbf{Q}_{t-K}^{(n)} \right) = \left[ \langle sg, Q, a \rangle_{t-K}^{(n)}; sg_t^{(n)}, Q_t^{(n)} \right]$. The $\square$ represents this predicted token neither participates in training nor inference so we ignore it. At the timestep $t$, different tokens are embedded by different linear layers and fed into the transformers (Vaswani et al., 2017) together. The output Q-function $\hat{Q}_t^{(n)}$ is processed by a linear layer.

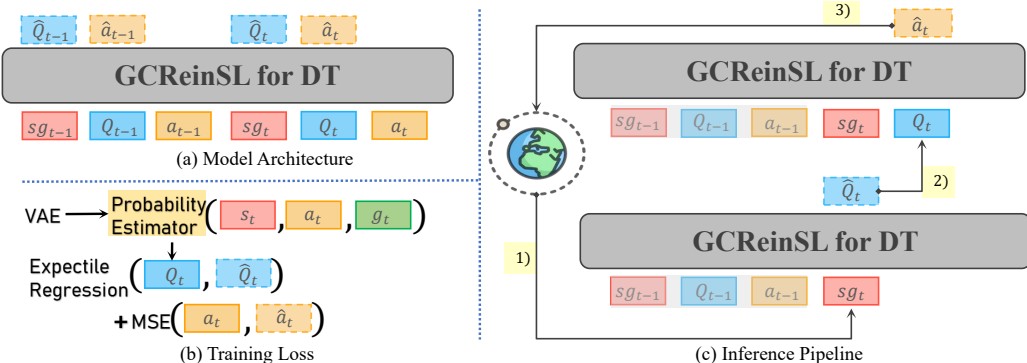

Figure 3: The overview of **GC*Rein*SL** for DT: (a) Model Architecture: The Q-function is the third inputs of **GC*Rein*SL** for DT and the outputs contain Q-value and actions. (b) Train Loss: As a Q-conditioned maximization sequence model, **GC*Rein*SL** for DT not only maximizes the action likelihood but also maximizes Q-function by expectile regression. (c) Inference Pipeline: When inference, **GC*Rein*SL** for DT first 1) gets state and goal from the environment to predict the in-distribution maximum Q-function. Then 2) predicted in-distribution max Q-function is concatenated with state and goal to predict the optimal action. Finally, 3) the environment executes the predicted action to Q-function the next state.

**Training Loss**  Since the model predicts two parts, $\hat{Q}_t$ and $\hat{a}_t$, the loss function is composed of Q-function loss and action loss. For the action loss, we adopt the MSE loss function of DT and simultaneously adjust the order of tokens:

$$\mathcal{L}_{\mathrm{a}} = \mathbb{E}_{t,n} \left[ a_t^{(n)} - \pi_\theta \left( \mathbf{SG}_{t-K}^{(n)}, \mathbf{Q}_{t-K}^{(n)} \right) \right]^2. \tag{10}$$

The Q-function loss is the expectile regression with the parameter $m$:

$$\mathcal{L}_{\mathrm{Q}}^m = \mathbb{E}_{t,n} \left[ |m - \mathbb{1}\left(\Delta Q < 0\right)| \Delta Q^2 \right], \tag{11}$$

$$\text{with } \Delta Q = Q_t^{(n)} - \pi_\theta \left( \mathbf{SG}_{t-K}^{(n)} \right).$$

Two loss functions have the same weight so the total loss is $\mathcal{L}_{\mathrm{a}} + \mathcal{L}_Q$.

**Inference Pipeline**  For each timestep $t$, the action is the last token, which means the predicted action is affected by state from the environment and the Q-function. The Q-function of the trajectories output by the sequence modeling exhibit a positive correlation with the initial conditioned Q-function (Chen et al., 2021; Zheng et al., 2022). That is, within a certain range, higher initial Q-function typically lead to better actions. In classical Q-learning (Mnih et al., 2015), the optimal value function $Q^*$ can derive the optimal action $a^*$ given the current state. In the context of sequence modeling, we also assume that the maximum Q-function are required to output the optimal actions. The inference pipeline of the **GC*Rein*SL** is summarized as follows:

$$\xrightarrow{\text{Env}} (sg_0) \xrightarrow{\pi_\theta} Q_0 \xrightarrow{\pi_\theta} a_0 \xrightarrow{\text{Env}} (sg_1) \xrightarrow{\pi_\theta} Q_1 \xrightarrow{\pi_\theta} a_1 \rightarrow \cdots \tag{12}$$

Specially, the environment initializes the state-goal pair $(sg_0)$ (i.e, $s_0$ and $g_0$ are concatenated to form $sg_0$) and then the sequence modeling $\pi_\theta$ predicts the maximum Q-function $Q_0$ given current state-goal pair $(sg_0)$. Concatenating $Q_0$ with $(sg_0)$, $\pi_\theta$ can output the optimal action $a_0$. Then the environment transitions to the next state $s_1$ and the reward $r_1$. It should be noted that this reward $r_1$ will **not** participate in the inference. Repeat the above steps until the trajectory comes to an end. The overall algorithm of **GC*Rein*SL** for DT is shown in Appendix B.1.

### 4.4.2 **GC*Rein*SL** FOR RvS

**Architecture**  To accommodate the Q-conditioned maximization for RvS (Emmons et al., 2021), which also predicts the maximum Q-function and utilizes it as a condition to guide the generation of optimal actions. Unlike **GC*Rein*SL** for DT, we construct a actor model for predicting actions and a

value model for predicting Q-function. In detail, the input of **GC*Rein*SL** for RvS and corresponding output are summarized as follows:

$$\textbf{Input:} \quad s_t, g_t, Q_t(s_t, a_t, g_t)$$

$$\textbf{Value Model Output:} \quad \hat{Q}_t(s_t, g_t)$$

$$\textbf{Actor Model Output:} \quad \hat{a}_t\left(s_t, g_t, \hat{Q}_t(s_t, g_t)\right)$$

When predicting the $\hat{Q}_t$, the value model takes the current state $s_t$ and desired goal $g_t$. For action $\hat{a}_t^{(n)}$, We adopt a actor model that incorporates Q-values for inference.

**Training Procedure and Inference Pipeline**   Like **GC*Rein*SL** for DT, the total loss function is also composed of Q-function loss and action loss, and the form is the same. At each step of the inference pipeline, the value model outputs the maximum Q-function value for the input state-goal pair, and then the actor model outputs the corresponding action. Note that in this state-goal pair, the state and the goal are treated as distinct elements. In the context of RvS, we also assume that the maximum Q-function are required to output the optimal actions. The training procedure is similar to that of **GC*Rein*SL** for DT, with the key distinction that the prediction of Q-values is generated by a value model. The inference pipeline of the **GC*Rein*SL** is summarized as follows:

$$\xrightarrow{\text{Env}} (s_0, g_0) \xrightarrow{v_\phi} Q_0 \xrightarrow{\pi_\theta} a_0 \xrightarrow{\text{Env}} (s_1, g_1) \xrightarrow{v_\phi} Q_1 \xrightarrow{\pi_\theta} a_1 \rightarrow \cdots \tag{13}$$

Specially, the environment initializes the state-goal pair $(s_0, g_0)$ and then the value model $v_\phi$ predicts the maximum Q-function $Q_0$ given current state-goal pair $(s_0, g_0)$. Concatenating $Q_0$ with $(s_0, g_0)$, $\pi_\theta$ can output the optimal action $a_0$. The overall algorithm of **GC*Rein*SL** for RvS is shown in Appendix B.2.

## 5 EXPERIMENTS

To rigorously evaluate the stitching capability of **GC*Rein*SL**, we employ the offline goal-conditioned datasets configuration as outlined in Ghugare et al. (2024). For the evaluation, we follow the methodology outlined by Ghugare et al. (2024), modifying the the **GC*Rein*SL** policy to navigate between previously unseen combinatorial (state, goal) pairs and subsequently measure the success rate. We then add the corresponding goal data augmentation techniques into the OCBC methods for a comparative analysis with our proposed approach. We additionally compared **GC*Rein*SL** with the previous sequence modeling methods on D4RL (Fu et al., 2020) complex offline `Antmaze-v2` datasets. Both offline goal-conditioned datasets are characterized by sparse rewards (i.e, reaching the goal results in a reward of 1, otherwise 0) and are designed to test stitching capabilities.

### 5.1 EXPERIMENTAL SETUP

We conducted a series of comparative experiments by implementing the OCBC methods within the same framework, as well as related goal data augmentation approaches. Specifically, we select RvS (Emmons et al., 2021) and DT (Chen et al., 2021), two competitive methods in OCBC, as baseline models for comparison. For goal data augmentation methods, we select Swapped Goal Data Augmentation (SGDA) (Yang et al., 2023) and Temporal Goal Data Augmentation (TGDA) (Ghugare et al., 2024) as advanced methodologies to serve as comparative baselines for our goal data augmentation study. SGDA (Yang et al., 2023) proposes a method that randomly choose augmented goals from different trajectories. TGDA (Ghugare et al., 2024) proposed a another goal data augmentation approach from the perspective of combinatorial optimization. It employs k-means (Lloyd, 1982) to cluster the goal and certain states into a group, and samples goals from later stages of these state trajectories as augmented goals. For related sequence modeling approaches, we select state-of-the-art methods, including Elastic Decision Transformer (EDT) (Wu et al., 2023) and Max-Return Sequence Modeling (Reinformer) (Zhuang et al., 2024), as baselines. Both of these methods, like ours, exhibit stitching property without requiring dynamic programming. Additionally, we compare these sequence modeling approaches to traditional reinforcement learning methods such as CQL and IQL. All experiments are conducted using five random seeds. Detailed implementations and hyperparameter settings are outlined in Appendix C and Appendix D, respectively.

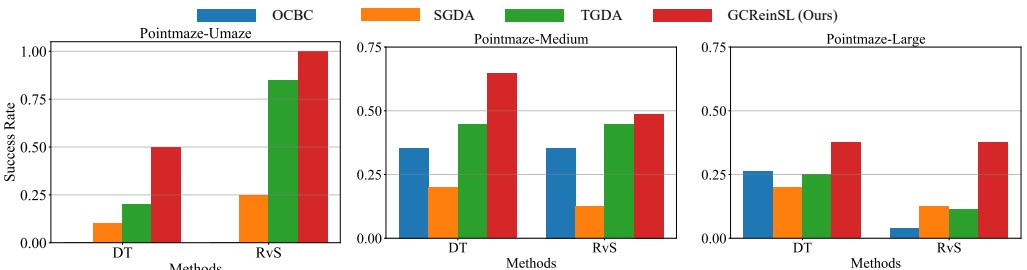

Figure 4: Performance of the original OCBC, as well as OCBC with corresponding goal data augmentation, compared to our SL method on the `Pointmaze` datasets from Ghugare et al. (2024). We use the final score as the report. **GC*Rein*SL** not only improves the performance of DT and RvS in all tasks, but also outperforms exist goal data augmentation methods.

## 5.2 TESTING THE ABILITY OF **GC*Rein*SL** AND COMPARED WITH PREVIOUS GOAL DATA AUGMENTATION METHODS

As shown in Fig. 4, it is evident that DT and RvS are struggle to demonstrate stitching property, particularly in the `Pointmaze-Umaze` and `Pointmaze-Large` datasets, where their performance is notably poor. However, when Q-conditioned maximization is incorporated into the OCBC methods, performance improvements were observed across all tasks, albeit to varying degrees. This enhancement is attributed to the fact that **GC*Rein*SL** allows for the sampling of unseen (state, goal) combinations during the training phase, thereby improving the generalization and stitching capability of the models. Our **GC*Rein*SL** consistently outperforms the other data augmentation approaches across all `Pointmaze` datasets, particularly in the more complex `Pointmaze-Medium` and `Pointmaze-Large` datasets. This suggests that our approach enables the selection of more suitable goals, facilitating more effective trajectory stitching.

## 5.3 SCALING TO HIGHER-DIMENSIONAL DATASETS

To evaluate the applicability of our **GC*Rein*SL** to tasks with higher-dimensional input spaces, we implemented it on a robotic control dataset with 111-dimensions (`Antmaze` (Ghugare et al., 2024)). In Fig. 5, we observe that **GC*Rein*SL** improves the performance of DT and RvS across all `Antmaze` datasets, with particularly notable improvements on the medium and large datasets.

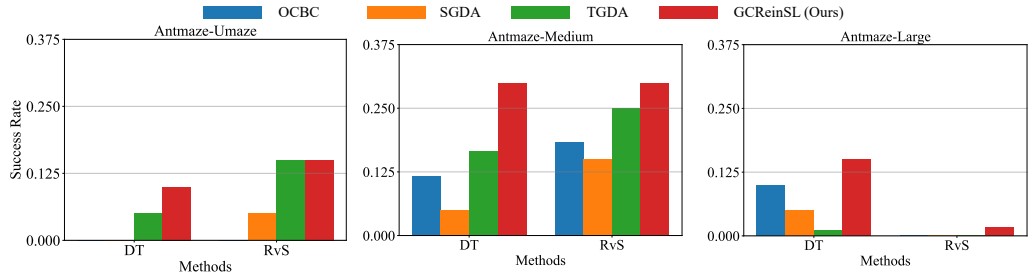

Figure 5: Performance on high-dimensional `Antmaze` datasets: **GC*Rein*SL** can still improve the performance of DT and RvS on high-dimensional `Antmaze` datasets. We also use the final score as the report. However, in some datasets such as `Antmaze-Medium`, **GC*Rein*SL** is inferior to advanced TGDA method.

## 5.4 COMPARED GC*Rein*SL WITH THE PREVIOUS MAX-RETURN SEQUENCE MODELING METHOD

We also compared our method with relevant sequence modeling approaches that perform stitching property on the standard offline dataset D4RL (Fu et al., 2020), specifically on the `Antmaze-v2` datasets, as shown in Table 1. From Table 1, it is evident that in the majority of the AntMaze datasets,

particularly in the complex medium and large AntMaze tasks, the **GC*Rein*SL** approach demonstrates superior performance, significantly closing the gap with TD learning methods such as CQL.

| Antmaze-v2 | RL | | Sequence Modeling | | | |
|---|---|---|---|---|---|---|
| | CQL | IQL | DT | EDT | Reinformer | **GC*Rein*SL (ours)** |
| umaze | **94.8 ± 0.8** | 84.00 ± 4.1 | 64.5 ± 2.1 | 67.8± 3.2 | 84.4±2.7 | 80.1±5.3 |
| umaze-diverse | 53.8 ± 2.1 | **79.5 ± 3.4** | 60.5 ± 2.3 | 58.3± 1.9 | 65.8±4.1 | 67.2±5.3 |
| medium-play | **80.5 ± 3.4** | 78.5 ± 3.8 | 0.8 ± 0.4 | 0.0± 0.0 | 13.2±6.1 | 49.0±3.5 |
| medium-diverse | 71.0 ± 4.5 | **83.5 ± 1.8** | 0.5 ± 0.5 | 0.0± 0.0 | 10.6±6.9 | 51.7±4.4 |
| large-play | 34.8 ± 5.9 | **53.5 ± 2.5** | 0.0 ± 0.0 | 0.6± 0.5 | 0.4 ±0.5 | 28.2±1.8 |
| large-diverse | 36.3 ± 3.3 | **53.0 ± 3.00** | 0.0 ± 0.0 | 0.0± 0.0 | 0.4 ±0.5 | 30.2±2.4 |
| *Total* | *371.2* | *432.0* | *126.3* | *126.7* | *174.8* | *306.4* |

Table 1: The normalized best score on D4RL (Fu et al., 2020) `Antmaze-v2` datasets. The results come from its original Reinformer (Zhuang et al., 2024) paper except **GC*Rein*SL**. The best result is **bold** and the blue result means the best result among sequence modeling.

## 5.5 ABLATION STUDY

In this section, we analyze the impact of the hyperparameter $L$ in the probability estimator and $m$ in the Q-function loss. As illustrated in the left panel of Fig. 6, the performance does not exhibit a linear relationship with increasing values of $L$. Therefore, we set $L = 500$ as the default value for the datasets employed in Ghugare et al. (2024). For the D4RL `Antmaze-v2` dataset (Fu et al., 2020), we select $L = 5$, in line with the methodology outlined by Wu et al. (2022).

As stated in Theorem 4.1, as $m \rightarrow 1$, the learned Q-function asymptotically converges to the maximum Q-function within the offline distribution.

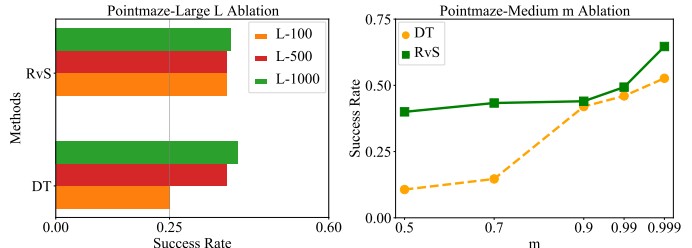

Given that a higher in-distribution Q-function corresponds to improved action selection, we can infer that performance will improve as $m$ approaches 1. The experimental results presented in the right panel of Fig. 6 are consistent with this theoretical prediction. However, larger values of $m$ do not consistently lead to more effective training or higher performance; in some cases, they may result in a performance decline. This could be attributed to overfitting to excessively large Q-function values present in the offline dataset.

Figure 6: Ablation study of different hyperparameter $L$ and $m$ in Ghugare et al. (2024) datasets. *(left)*: The performance on the `Pointmaze-Large` dataset when applying different values of $L$ to the importance sampling estimator. *(right)*: The trend of last results as $m$ varies on `Pointmaze-Medium` dataset.

## 6 CONCLUSION

In this work, we propose the paradigm of Q-conditioned maximization supervised learning which considers the RL objective that maximizes Q-function for SL-based methods (OCBC methods). Both theoretical analysis and experiments indicate that our proposed model **GC*Rein*SL** reduces the performance gap between itself and classical RL approaches. However, our approach still exhibits a gap compared to classical RL methods and is sensitive to certain hyperparameters. Future work could focus on developing more robust SL architectures that are better suited for scenarios where classical RL excels, particularly in trajectory stitching. This would provide a more nuanced understanding of the respective strengths and applications of each approach.

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

# Contents of Appendix

# A    PROOFS

In this section, we restate theorems in the paper and present their proofs.

## A.1    PROOF OF THEOREM 3.1

**Definitions**    Before proving this theorem, we first have the following definitions:

(1) We begin by defining the Q-function in the form of the expected reward:

$$Q^\pi(s, a, g) \triangleq \mathbb{E}_{\pi(\cdot|g)} \left[ \sum_{t=0}^\infty \gamma^t r(s_t, a_t, g) \mid {}^{s_0=s,}_{a_0=a} \right]. \tag{14}$$

(2) Then we will define rewards conditioned with goal $g$ as:

$$r(s, a, g) \triangleq \begin{cases} (1-\gamma)\big(p_0(s_0 = g) + \gamma p(s_1 = g \mid s_0, a_0)\big), & t = 0 \\ (1-\gamma)\gamma p(s_{t+1} = g \mid s_t, a_t), & t > 0. \end{cases} \tag{15}$$

(3) Finally, We define the discounted state occupancy distribution, as:

$$p_+^\pi(g) = (1-\gamma) \sum_{t=0}^\infty \gamma^t p_t^\pi(g). \tag{16}$$

And We can rewrite Eq. (16) as

$$p_+^\pi(g) = (1-\gamma)p_0^\pi(g) + (1-\gamma) \sum_{t=1}^\infty \gamma^t p_t^\pi(g). \tag{17}$$

**Proof Objective**    Our objective is to establish a relationship between the Q-function and the discounted state occupancy distribution:

$$Q^\pi(s, a, g) = p_+^\pi(g \mid s, a) \tag{18}$$

**Proof**    We begin by examining the term for $t = 0$, followed by an analysis of the term for $t > 0$. The probability of visiting a state at time $t = 0$ corresponds to the initial state distribution:

$$p_0^\pi(g) = p_0(g).$$

For $t > 0$, the term $p_t^\pi(g)$ in Eq. (17) is a probability of reaching the goal $g$ at timestep $t$ with policy conditioned on $g$, then we can write this term as follows:

$$\begin{aligned} p_t^\pi(g) &= \mathbb{E}_{\pi(\cdot|g)} \left[ p_t(g \mid s_{t-1}, a_{t-1}) \right] \\ &= \mathbb{E}_{\pi(\cdot|g)} \left[ p(s_t = g \mid s_{t-1}, a_{t-1}) \right]. \end{aligned}$$

In the second line, we apply the Markov property, which implies that the probability of reaching $g$ at time $t$ depends solely on the dynamics, $p(s_{t+1} \mid s_t, a_t)$.

Substituting this into Eq. (17), we obtain:

$$p_+^\pi(g) = (1-\gamma)p_0^\pi(g) + (1-\gamma)\sum_{t=1}^{\infty}\gamma^t p_t^\pi(g)$$

$$= (1-\gamma)p_0^\pi(g) + (1-\gamma)\sum_{t=1}^{\infty}\gamma^t\mathbb{E}_{\pi(\cdot|g)}\big[p(s_t = g \mid s_{t-1}, a_{t-1})\big]$$

$$= (1-\gamma)p_0^\pi(g) + (1-\gamma)\sum_{t=0}^{\infty}\gamma^{t+1}\mathbb{E}_{\pi(\cdot|g)}\big[p(s_{t+1} = g \mid s_t, a_t)\big]$$

$$= (1-\gamma)p_0^\pi(g) + (1-\gamma)\mathbb{E}_{\pi(\cdot|g)}\left[\sum_{t=0}^{\infty}\gamma^{t+1}p(s_{t+1} = g \mid s_t, a_t)\right]$$

$$= \mathbb{E}_{\pi(\cdot|g)}\left[(1-\gamma)p_0(s_0 = g) + (1-\gamma)\sum_{t=0}^{\infty}\gamma^{t+1}p(s_{t+1} = g \mid s_t, a_t)\right]$$

$$= \mathbb{E}_{\pi(\cdot|g)}\left[\underbrace{(1-\gamma)\left(p_0(s_0 = g) + \gamma p(s_1 = g \mid s_0, a_0)\right)}_{r(s_0, a_0, g)} + \sum_{t=1}^{\infty}\gamma^t \underbrace{(1-\gamma)\gamma p(s_{t+1} = g \mid s_t, a_t)}_{r(s_t, a_t, g)}\right]$$

$$= \mathbb{E}_{\pi(\cdot|g)}\left[\sum_{t=0}^{\infty}\gamma^t r(s_t, a_t, g)\right].$$

In the third line, we adjust the bounds of the summation to begin at 0, modifying the terms inside the summation accordingly. In the fourth line, we apply the linearity of expectation to shift the summation inside the expectation. In the fifth line, we again utilize the linearity of expectation to incorporate the term for $t = 0$ within the expectation. In the final two lines, we substitute the definition of $r(s, a, g)$ to derive the desired result.

For a set state-action pair $(s, a)$, we can obtain:

$$p_+^\pi(g \mid s, a) = \mathbb{E}_{\pi(\cdot|g)}\left[\sum_{t=0}^{\infty}\gamma^t r(s_t, a_t, g) \mid {}_{a_0=a}^{s_0=s}\right] = Q^\pi(s, a, g). \tag{19}$$

Thus, the relationship between the Q-function and the discounted state occupancy distribution is formally established.

## A.2 PROOF OF THEOREM 4.1

**Definitions**   Before proving this theorem, we first have the following definitions:

(1) Expectile Regression Loss: The $m$-expectile regression loss for a predicted Q-function $\mathbf{Q}^m$ ($\mathbf{Q}^m := \mathbf{Q}^m(\mathbf{SG}) = \arg\min \mathcal{L}_Q^m(\mathbf{SG})$, $\mathbf{SG} := (s, g, a, Q)$):

$$\mathcal{L}_Q^m = \mathbb{E}_{(s,a,g)\in\mathcal{D}}\left[|m - \mathbb{1}(\Delta Q < 0)|\,\Delta Q^2\right], \tag{20}$$

here $Q = Q^\pi(s, a, g)$, $\Delta Q = Q - \mathbf{Q}^m$ and $\mathbf{Q}^m$ can come from the supervised learning model. $\mathbb{1}(\Delta Q < 0)$ is an indicator function that equals 1 when $(\Delta Q < 0)$. This loss introduces an asymmetric penalty depending on whether $\mathbf{Q}^m$ overestimates or underestimates the target $Q(s, a, g)$.

(2) Maximum Q-function: The maximum Q-function with actions for a given $(s, a, g)$ from offline dataset $\mathcal{D}$:

$$Q_{\max} = \max_{\mathbf{a}\sim\mathcal{D}} Q(s, a, g) \tag{21}$$

Note that $Q(s, a, g)$ is estimated from the offline dataset $\mathcal{D}$ using a VAE model, as detailed in Section 4.2.

(3) Element-wise Interpretation: All inequalities involving $\mathbf{Q}^m$ in this proof are interpreted element-wise, meaning they apply independently to each tuple $(s, a, g)$ in the offline dataset.

**Proof Objective** Suppose the Q-function is predicted by the supervised learning model itself using $m$-expectile regression, For $m \in (0,1)$, let this predicted Q-function be $\mathbf{Q}^m$, which minimizes the expectile regression loss $\mathcal{L}_Q^m$. Then as $m \to 1$, $\mathbf{Q}^m \to Q_{max}$.

**Proof** The proof primarily relies on the monotonicity property of $m$-expectile regression and employs a proof by contradiction.

Firstly, leveraging the monotonicity property of $m$-expectile regression (Newey & Powell, 1987), it follows that $\mathbf{Q}^{m_1} \le \mathbf{Q}^{m_2}$ for $0 < m_1 < m_2 < 1$.

Secondly, for all $m \in (0,1)$, it holds that $\mathbf{Q}^m \le Q_{\max}$. Assume there exists some $m_3$ such that $\mathbf{Q}^{m_3} > Q_{\max}$. In this case, all Q-values from the offline dataset would satisfy $Q < \mathbf{Q}^{m_3}$. Consequently, the Q-function loss can be simplified, given the constant weight $1 - m_3$.

$$
\begin{aligned}
\mathcal{L}_{\mathbf{Q}}^{m_3} &= \mathbb{E}\left[(1-m_3)\left(Q - \mathbf{Q}^{m_3}\right)^2\right] \\
&> \mathbb{E}\left[(1-m_3)\left(Q - \max[Q_t^{(n)}]\right)^2\right].
\end{aligned}
$$

This inequality holds because $Q \le \max[Q] < \mathbf{Q}^{m_3}$. However, this contradicts the fact that $\mathbf{Q}^{m_3}$ is derived by minimizing the Q-function loss. Therefore, the assumption is invalid, and we conclude that $\mathbf{Q}^m \le Q_{\max}$ is true. This proof step demonstrates that the predicted Q-function does not suffer from out-of-distribution (OOD) issues.

Finally, the convergence to this limit is a direct consequence of the properties of bounded and monotonically non-decreasing functions, thereby demonstrating the validity of the theorem.

A.3 PROOF OF COROLLARY 1

The conclusion drawn from Furuta et al. (2021) indicates that the OCBC methods can be summarized as performing **Hindsight Information Matching (HIM)**: Given a offline dataset $\mathcal{D}$ and its information statistics $I(\tau_t)$, OCBC methods are trying to learn a goal-conditioned policy $\pi(a|s,g)$ whose trajectory rollouts satisfy some desired information statistics value $g$:

$$
\min_\pi E_{g \sim \mathcal{D}}\left[D(I(\tau), g)\right], \tag{22}
$$

where $D$ is a divergence measure for information matching such as Kullback-Leibler (KL) divergence. Within the **HIM** framework, the optimization objective of Q-conditioned maximization supervised learning can be interpreted as aligning with the statistical property of future trajectories. In goal-conditioned reinforcement learning (RL), this statistical information is defined as the probability of reaching the goal $g$ in the future. Since the Q-function aggregates future rewards, it acts as a statistical summary of the trajectory $\tau_1$ (i.e., the expected maximum return). Therefore, the Q-value in Q-conditioned maximization supervised learning can be understood as the trajectory information statistic $I(\tau)$ within the HIM framework:

$$
I(\tau) = Q^\pi(s, a, g). \tag{23}
$$

Thus, the optimization objective of Q-conditioned maximization supervised learning can be expressed as:

$$
\min_\pi E_{g \sim \mathcal{D}}\left[D(Q^\pi(s, a, g), g)\right]. \tag{24}
$$

This is equivalent to the **HIM** objective of aligning trajectory statistics with a defined statistical objective. Both approaches optimize the policy by matching the future trajectory information to the desired objective. Consider two trajectories in the offline dataset: $\tau_1 = \left\{< s_0^1, a_0^1, r_0^1 >, < s_1^1, a_1^1, r_1^1 >, ..., < s_T^1, a_T^1, r_T^1 >\right\}$ and $\tau_2 = \left\{< s_0^2, a_0^2, r_0^2 >, < s_2^2, a_2^2, r_2^2 >, ..., < s_T^2, a_T^2, r_T^2 >\right\}$, which respectively reach goals $g_1$ and $g_2$. If we start from state $s_0^1$ and expect to reach the final goal $g$, but the goal $g_1$ achieves a lower cumulative reward compared to the reached goal $g_2$, Q-conditioned maximization supervised learning will tend to select $g_2$ as the global goal. Consequently, $g_2$ can be utilized as an augmented goal for the initial state $s_0^1$, enhancing the overall trajectory performance. In summary, Q-conditioned maximization supervised learning attains the optimal policy by selecting high-reward goals and stitching together distinct trajectory segments.

# B GC*Rein*SL ALGORITHM

Below we provide a detailed outline of the **GC*Rein*SL** algorithm for DT and RvS.

## B.1 GC*Rein*SL ALGORITHM FOR DT

---
**Algorithm 1 GC*Rein*SL for DT**

---
1: **Input:** offline dataset $\mathcal{D}$, sequence modeling $\pi_\theta$
2: Initialize VAE with parameters $\psi$ and $\varphi$
3: **Function** VAE Training
4:     Sample minibatch of transitions from offline dataset $\mathcal{D}$: $(s, a, g) \sim \mathcal{D}$
5:     Update $\psi, \varphi$ minimizing $\mathcal{L}_{\text{ELBO}}(s, a, g; \varphi, \psi)$ in Eq. (7)
6: //Training Procedure
7: **for** sample $\langle \cdots, s_t, g_t, a_t \rangle$ from $\mathcal{D}$ **do**
8:     Get $Q_t$ with probability estimator with Eq. (8)
9:     Get $\hat{Q}_t, \hat{a}_t$ with sequence modeling $\pi_\theta$: $\hat{Q}_t, \hat{a}_t = \pi_\theta(\cdots, sg_t, a_t, Q_t)$
10:     Calculate total loss $\mathcal{L}_a + \mathcal{L}_Q^m$ by Equation Eq. (10) and Eq. (11)
11:     Take gradient descent step on $\nabla_\theta \left( \mathcal{L}_a + \mathcal{L}_Q^m \right)$
12: **end for**
13: //Inference Pipeline
14: **Input:** sequence modeling $\pi_\theta$, environment Env
15: $s_0 = \text{Env}.reset(\ )$ and $t = 0$
16: **repeat**
17:     Predict maximum Q-function $\hat{Q}_t = \pi_\theta(\cdots, sg_t, \square, \square\ )$
18:     Predict optimal action $\hat{a}_t = \pi_\theta\left(\cdots, sg_t, \hat{Q}_t, \square\right)$
19:     $s_{t+1}, r_t = \text{Env}.step(\hat{a}_t)$ and $t = t + 1$
20: **until** done

---

## B.2 GC*Rein*SL ALGORITHM FOR RvS

---
**Algorithm 2 GC*Rein*SL for RvS**

---
1: **Input:** offline dataset $\mathcal{D}$, actor model $\pi_\theta$, value model $v_\phi$
2: VAE training is similar to **GC*Rein*SL** for DT.
3: //Training Procedure
4: **for** sample $\langle \cdots, s_t, g_t, a_t \rangle$ from $\mathcal{D}$ **do**
5:     Get $Q_t$ with probability estimator with Eq. (8)
6:     Predict maximum Q-function $\hat{Q}_t = v_\phi(s_t, g_t)$
7:     Predict optimal action $\hat{a}_t = \pi_\theta\left(s_t, g_t, \hat{Q}_t\right)$
8:     The calculation of the total loss is also the same as in **GC*Rein*SL** for DT.
9: **end for**
10: //Inference Pipeline
11: **Input:** value model $v_\phi$, actor model $\pi_\theta$, environment Env
12: $s_0 = \text{Env}.reset(\ )$ and $t = 0$
13: **repeat**
14:     Predict maximum Q-function $\hat{Q}_t = v_\phi(s_t, g_t)$
15:     Predict optimal action $\hat{a}_t = \pi_\theta\left(s_t, g_t, \hat{Q}_t\right)$
16:     $s_{t+1}, r_t = \text{Env}.step(\hat{a}_t)$ and $t = t + 1$
17: **until** done

---

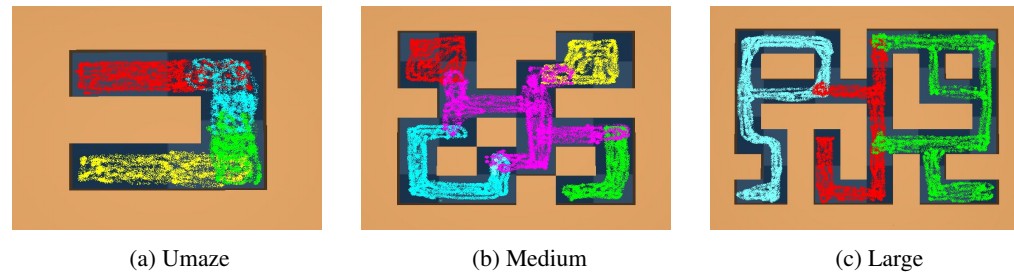

|             |              |             |
|:-----------:|:------------:|:-----------:|
| (a) Umaze   | (b) Medium   | (c) Large   |

Figure 7: Goal-conditioned datasets from Ghugare et al. (2024): Different colors represent the navigation regions of various data collection policies. During data collection, these policies navigate between randomly selected state-goal pairs within their respective navigation regions. These visualizations pertain to the `Pointmaze` dataset, with similar patterns observed in the `Antmaze` dataset.

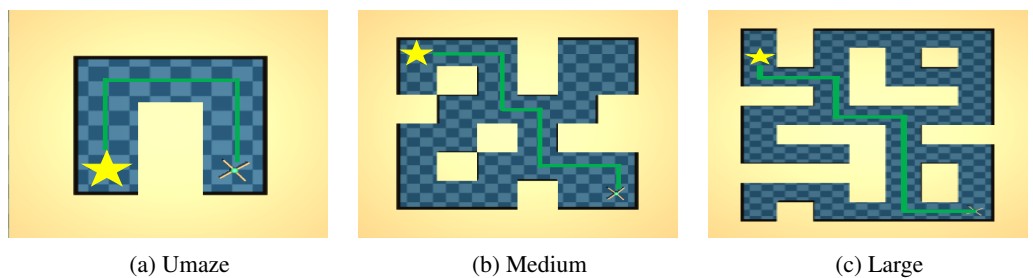

|             |              |             |
|:-----------:|:------------:|:-----------:|
| (a) Umaze   | (b) Medium   | (c) Large   |

Figure 8: Goal-conditioned Datasets from Fu et al. (2020): The `AntMaze-v2` datasets involve controlling an 8-DoF quadruped to navigate towards a specified goal state. This benchmark requires value propagation to effectively stitch together sub-optimal trajectories from the collected data.

## C    EXPERIMENT DETAILS

In this section we provide all the implementation details as well as hyperparameters used for all the algorithms in our experiments – DT, RvS, VAE, and **GC*Rein*SL**.

### C.1    OFFLINE DATASETS

**Goal-conditioned Datsets from Ghugare et al. (2024)**    We utilize the `Pointmaze` and `Antmaze` datasets, as presented in Ghugare et al. (2024). As described in Section 5, both offline datasets contain $10^6$ transitions and are specifically constructed to evaluate trajectory stitching in a combinatorial setting (see Fig. 7). In the `Pointmaze` dataset, the task involves controlling a ball with two degrees of freedom by applying forces along the Cartesian x and y axes. By contrast, the `Antmaze` dataset features a 3D ant agent, provided by the Farama Foundation (Towers et al., 2023). The `Pointmaze` datasets were collected using a PID controller, while the `Antmaze` datasets were generated using a pre-trained policy from D4RL (Fu et al., 2020). Visual representations of the various `Pointmaze` configurations can be found in Fig. 7.

**Goal-conditioned Datasets from Fu et al. (2020)**    In the experiments comparing with related sequence modeling approaches, we follow the methodology outlined in Zhuang et al. (2024) to construct the `AntMaze-v2` datasets using D4RL, which also contain $10^6$ transitions (see Fig. 8). These `AntMaze-v2` datasets are characterized by sparse rewards, where $r = 1$ is awarded upon reaching the goal. Both the medium and large datasets lack complete trajectories from the starting point to the goal, requiring the algorithm to stitch together incomplete or failed trajectories to achieve the desired goal.

## C.2 IMPLEMENTATION DETAILS

We use the default configurations of DT and RvS as described in Ghugare et al. (2024), with some values modified. Note that in specific datasets, certain parameter values have been adjusted. The architecture and training process of the VAE are identical to those described in SPOT (Wu et al., 2022).

Our **GC*Rein*SL** for DT implementation draws inspiration from and references the following four repositories:

- TGDA: https://github.com/RajGhugare19/stitching-is-combinatorial-generalisation;
- SPOT: https://github.com/thuml/SPOT;
- Reinformer: https://github.com/Dragon-Zhuang/Reinformer.

The state-goal pair tokens, Q-function tokens and action tokens are first processed by different linear layers. Then these tokens are fed into the decoder layer to obtain the embedding. Here the decoder layer is a lightweight implementation from Reinformer (Zhuang et al., 2024). The context length for the decoder layer is denoted as $K$. Our **GC*Rein*SL** for RvS implementation is similar to the idea of **GC*Rein*SL** for DT, but it is divided into value networks and policy networks. The value network outputs the expected Q-function from state $s$ to goal $g$. This expected Q-function, along with the state $s$ and goal $g$, is then used as input to the policy network. We employed both the AdamW (Loshchilov, 2017) and Adam (Kingma & Ba, 2014) optimizers to optimize the total loss (i.e, action loss and Q-function loss) for DT and RvS, respectively, in alignment with the methodologies outlined in their original papers. The hyperparameter of Q-function loss is denoted as $m$.

## D HYPERPARAMETERS

In this section, we will provide a detailed description of parameter settings for in our experiments. The hyperparameters of SGDA and TGDA remain consistent with their original settings. For fair comparison, our method still sets the same augmentation rate of 0.5 as theirs. The hyperparameters of **GC*Rein*SL** for DT in various datasets are presented in the tables below. In all tables, the arrows indicate the directional change in the corresponding values for RvS.

### D.1 HYPERPARAMETER $m$

The hyperparameter $m$ is crucially related to the Q-function loss and is one of our primary focuses for tuning. We explore values within the range of $m = [0.7, 0.9, 0.99, 0.999]$. When $m = 0.5$, the expectile loss function will degenerate into MSE loss, which means the model is unable to output a maximized Q-function. So we do not take $m = 0.5$ into consideration. We observe that performance is generally lower at $m = 0.9$ compared to others except Pointmaze-Umaze. Only Pointmaze-Large adopt the parameter $m = 0.999$ while $m = 0.99$ are generally better than $m = 0.999$ on other datasets. The detailed hyperparameter selection of $m$ is summarized in the following table:

Table 2: Hyperparameters $m$ of Q-function loss on different datasets.

| Dataset | $m$ | | |
|---|---|---|---|
| | | Antmaze-Umaze | 0.9 |
| Pointmaze-Umaze | $0.99 \to 0.9$ | Antmaze-umaze-diverse | 0.99 |
| Pointmaze-Medium | 0.99 | Antmaze-medium-play | 0.99 |
| Pointmaze-Large | $0.99 \to 0.999$ | Antmaze-medium-diverse | 0.99 |
| Antmaze-Umaze | 0.99 | Antmaze-large-play | 0.99 |
| Antmaze-Medium/Large | 0.99 | Antmaze-large-diverse | 0.99 |

## D.2 CONTEXT LENGTH $K$

The context length $K$ is another key hyperparameter in **GC*Rein*SL** for DT, and we conduct a parameter search across the values $K = [2, 5, 10, 20]$. The maximum value is 20 because the default context length for DT (Chen et al., 2021) is 20. The minimum is 2, which corresponds to the shortest sequence length (setting $K = 1$ would no longer constitute sequence learning). Overall, we found that $K = 10$ and $K = 20$ lead to more stable learning and better performance on Ghugare et al. (2024) `Pointmaze` and `Antmaze` datasets. Conversely, a smaller context length is preferable on D4RL (Fu et al., 2020) `Antmaze-v2` dataset. The parameter $K$ has been summarized as follows:

Table 3: Context length $K$ on different datasets.

| Dataset | $K$ | | |
|---|---|---|---|
| | | Antmaze-Umaze | 2 |
| Pointmaze-Umaze | 10 | Antmaze-umaze-diverse | 2 |
| Pointmaze-Medium | 10 | Antmaze-medium-play | 3 |
| Pointmaze-Large | 5 | Antmaze-medium-diverse | 2 |
| Antmaze-Umaze | 20 | Antmaze-large-play | 3 |
| Antmaze-Medium/Large | 20 | Antmaze-large-diverse | 2 |

## D.3 TRAINING STEPS AND LEARNING RATE

The default number of training steps is 50000, with a learning rate of 0.0002. With these default settings, if the training score continues to rise, we would consider increasing the number of training steps or doubling the learning rate. For some datasets, 50000 steps may cause overfitting and less training steps are better. The training steps are presented in Table 4. The learning rate remains unchanged across all (Ghugare et al., 2024) goal-conditioned datasets and is set to be the same on the goal-conditioned dataset (Fu et al., 2020) as in (Zhuang et al., 2024). We evaluate the policy every 10 times to obtain a mean success rate in goal-conditioned datasets or normalized score in goal-conditioned datasets. For each seed, the mean success rate and normalized score are all calculated as the average results of 100 trajectories.

Table 4: The training steps on different datasets.

| Dataset | Training Steps | | |
|---|---|---|---|
| | | Antmaze-umaze | 100000 |
| Pointmaze-Umaze | $50000 \rightarrow 18000$ | Antmaze-umaze-diverse | 50000 |
| Pointmaze-Medium | $80000 \rightarrow 30000$ | Antmaze-medium-play | 100000 |
| Pointmaze-Large | $80000 \rightarrow 50000$ | Antmaze-medium-diverse | 100000 |
| Antmaze-Umaze | $50000 \rightarrow 60000$ | Antmaze-large-play | 100000 |
| Antmaze-Medium/Large | $80000 \rightarrow 100000$ | Antmaze-large-diverse | 100000 |

# E TRAINING CURVES

We exhibit the training curves on five seeds. The black line represents the mean of these five seeds and the red shaded area represents the variance.

## E.1 GOAL-CONDITIONED DATASETS FROM GHUGARE ET AL. (2024)

The training curves for nine datasets from Ghugare et al. (2024) are shown in Fig. 10. The training process for `Pointmaze-Umaze` exhibits relatively stable behavior. However, the training on `Pointmaze-Medium` and `Pointmaze-Large` is characterized by high variance and significant fluctuations. Similarly, the `Antmaze-Umaze` dataset shows some degree of instability, while the performance on the `Antmaze-Medium` dataset is particularly poor.

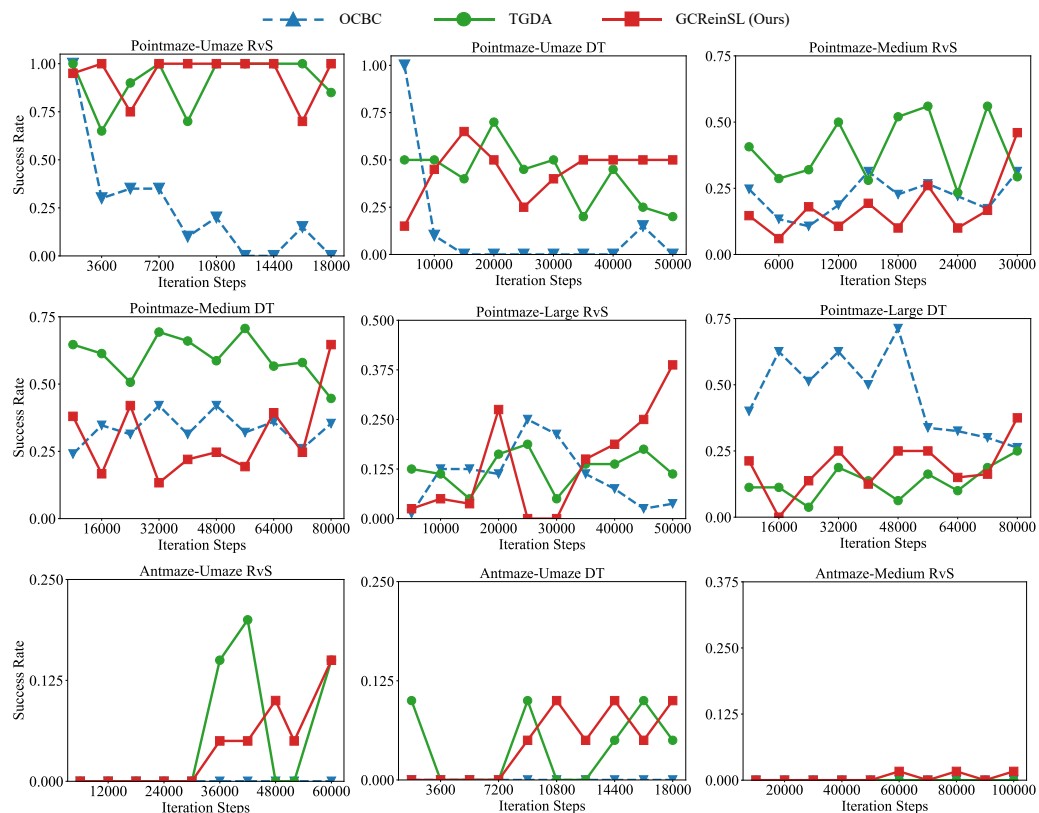

Figure 9: Training curves of OCBC and related goal data augmentation methods on Ghugare et al. (2024) dataset. Although our **GC*Rein*SL** method exhibits some instability on certain datasets, on average, **GC*Rein*SL** tends to improve and achieves promising results with extended training. A potential direction for future research is to develop a more robust **GC*Rein*SL** method that requires less hyperparameter tuning.

## E.2 GOAL-CONDITIONED DATASETS FROM FU ET AL. (2020)

Since we report the best score during training rather than the final score, we do not include training curves for `Antmaze`. As the `Antmaze` datasets contain sparse rewards, to prevent the occurrence of invalid values during training, we follow the approach of Zhuang et al. (2024) and modify the reward function to $\hat{r} = 100 \times r + 1$. In the Fig. 10, we visualize the performance of the state-of-the-art Reinformer algorithm and our method on `Antmaze`, and compare the results with those of the classic TD learning algorithm, IQL. In Fig. 10, we provide a detailed performance comparison with TD learning methods.

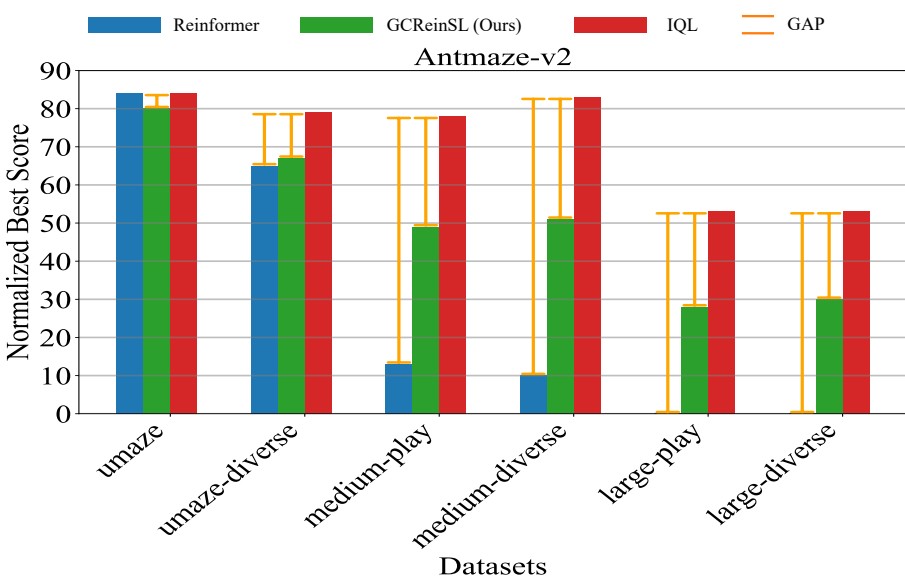

Figure 10: Performance of Reinfromer and **GC*Rein*SL** on four different goal-conditioned `Antmaze-v2` datasets from Fu et al. (2020). The gap between the two orange bars represents the difference from the IQL algorithm, with shorter gaps indicating better performance. Our SL method outperforms advanced method Reinformer across three datasets, further reducing the gap with TD learning methods.

