# OpenReview forum: "Bridging the Gap Beteween SL and TD Learning via Q-conditioned maximization"
_ICLR.cc/2025/Conference — Submitted to ICLR 2025_

### Official Review · Reviewer_zjNR · 2024-11-01

**Soundness:** 2
**Presentation:** 2
**Contribution:** 2
**Rating:** 3
**Confidence:** 3

**Summary:**

This paper studies reinforcement learning via surpervised learning and explores how to endow SL with trajectory stitching ability. Goal-Conditioned Reinforced Supervised Learning (GCReinSL) is proposed which emphasizes the maximization of the Q-function during the training phase to estimate the maximum expected return within the distribution, subsequently guiding optimal action selection during the inference process.

**Strengths:**

The paper is relatively well-written. Experiment results are solid.

**Weaknesses:**

The idea is not novel, basically a combination of tricks in exsiting literature. The expriments results can not support the title. See my questions for more details.

**Questions:**

1. It might be confusing to use $\pi$ to denote the probabiliy over the trajectory in $\pi(\tau|s,g)$ (3) and also to denote the policy $\pi(a|s,g)$.

2. In section 4.3, what is the $\pi$ in the probability distribution? at first, it was $p^{\pi}$ in line 234, then it becomes  $p^{\pi(\cdot|\cdot|g)}$ in line 245. Is it the behavior policy collecting the offline dataset?

3. For the Antmaze taks and the results in Table 1. The DT, EDT and Reinformer almost do not work. GCReinSL improves the performance from approximately 0 to about 10 (with large variance), there is a huge gap compared to RL method (about 50-80). Say, the improvement is about 10, and the initial gap is 50-80. Is it proper to claim 'significantly narrowing the gap with TD learning methods such as IQL'? I see this experiment as a example that SL would fail catastrophically, even with your maximum Q conditioning trick.

4. I have a question about the maximum Q conditioning trick. Different from the Return conditioned supervised learning methods such as Reinformer, for which they can directly access to the return in the dataset, in the goal conditioned supervised learning, the Q-function is estimated from VAE, and then the maximization is performed on the estimated Q-function. I guess the estimation error is hard to control as it may come from multiple sources: 1) how you evaluate that the VAE obtain decent estimation of the goal probability? 2) how you sure that the expectile regression gives a proper maximum in distribution Q value? Theorem 4.1 is not a accurate quantification of the return you get as it only considers the ideal case m goes to 1 and it does not consider how the maximum value is cover in the offline dataset.

---

> ### Author Response · Authors · 2024-12-02
> **Author Responses**
>
> Thank you for the detailed review and for the suggestions for improving the work. Below, I have carefully addressed your comments and concerns.
>
> For Question 1 and Question 2:
>
> We have addressed these two issues, as detailed in the revised version under Section 3.1 and Section 4.3.
>
> For Question 3:
>
> First, sequence modeling methods such as DT, EDT, and Reinformer exhibit limited stitching capabilities. Second, after carefully reviewing the code, we found that the limited improvement over IQL might be due to using the original parameter settings of Reinformer in the goal-conditioned RL context. To address this, we adjusted the training steps, learning rate, and sequence length to better suit the goal-conditioned RL context. As shown in the revised version's Table 1, GCReinSL for DT significantly improves the stitching capability of sequence modeling (with a 230-point increase compared to Reinformer), substantially narrowing the gap with IQL.
>
> For Question 4:
>
> It is important to note that this context pertains to goal-conditioned RL (as clarified in the revised version, it is explicitly not a return-conditioned RL setting). Therefore, leveraging the relationship between the Q-function and probabilities described in Theorem 3.1, we first estimate probabilities using a VAE and then maximize the Q-function. In a return-conditioned RL setting, the returns from the offline dataset can be directly utilized instead. Notably, using a VAE as a probabilities estimator is a well-established approach, as demonstrated in works such as [1], [2], and [3].
>
> [1]Fujimoto, Scott, David Meger, and Doina Precup. "Off-policy deep reinforcement learning without exploration." ICML, 2019.
>
> [1] Zhou, Wenxuan, Sujay Bajracharya, and David Held. "Plas: Latent action space for offline reinforcement learning." CoRL, 2021.
>
> [1] Wu, Jialong, et al. "Supported policy optimization for offline reinforcement learning." NIPS, 2022.

---

### Official Review · Reviewer_KRBU · 2024-11-02

**Soundness:** 2
**Presentation:** 2
**Contribution:** 2
**Rating:** 5
**Confidence:** 4

**Summary:**

This paper studies the stitching property for SL within goal-conditioned offline RL problems. This stitching property is commonly obtained in TD-based algorithms and fails in SL. This paper proposes the GCReinSL, which enhances SL-based approaches by incorporating maximize Q-function.  Equipped with the GCReinSL framework, the previous outcome-conditioned behavioral cloning (OCBC) algorithms exhibit the switching property and achieve better performance under the goal-conditioned setting.

**Strengths:**

1. The difference in trajectory stitching property between SL and TD does exist and is valuable for studying to improve the generalization performance of SL.
2.  The method that incorporates maximizing Q-function is natural, and the experiments show effectiveness.

**Weaknesses:**

1. The introduction lacks sufficient emphasis on motivation, such as the advantages and necessity of SL compared to TD under the goal-conditional setting. It would be better to discuss the importance of SL-based methods, like OCBC, in detail in the introduction.
2. Following Weakness 1, the experimental results also show a significant gap compared to TD-based algorithms (Table 1). It is still helpful to discuss this experiment phenomenon after Table 1.

**Questions:**

Please see the Weakness part.
1. As shown in Figure 5, the performance of GCReinSL is inferior to the advanced TGDA method in some higher-dimensional tasks. Could the author discuss this result in detail?

---

> ### Author Response · Authors · 2024-12-02
> **Author Responses**
>
> We thank the reviewer for the detailed review. Below, we address the raised concerns and questions.
>
> 1. The introduction lacks sufficient emphasis on motivation, such as the advantages and necessity of SL compared to TD under the goal-conditional setting. It would be better to discuss the importance of SL-based methods, like OCBC, in detail in the introduction.
>
> Thank you for your valuable suggestions on the paper. We have revised the introduction based on your feedback, providing a detailed discussion on the importance of SL-based methods, such as OCBC. The specific changes can be found in lines 50-57 of the revised version.
>
> 2. Following Weakness 1, the experimental results also show a significant gap compared to TD-based algorithms (Table 1). It is still helpful to discuss this experiment phenomenon after Table 1.  and 3. As shown in Figure 5, the performance of GCReinSL is inferior to the advanced TGDA method in some higher-dimensional tasks. Could the author discuss this result in detail?
>
> First, sequence modeling methods such as DT, EDT, and Reinformer exhibit limited stitching capabilities. Second, after carefully reviewing the code, we found that the limited improvement over IQL might be due to using the original parameter settings of Reinformer in the goal-conditioned RL context. To address this, we adjusted the training steps, learning rate, and sequence length to better suit the goal-conditioned RL context. As shown in the revised version's Table 1, GCReinSL for DT significantly improves the stitching capability of sequence modeling (with a 230-point increase compared to Reinformer), substantially narrowing the gap with IQL.

---

> > ### Comment · Reviewer_KRBU · 2024-12-03
> >
> > Thanks for the author's reply and additional experiments. By carefully choosing the parameters of GCReinSL, the revision paper improves the performance of the proposed method.
> >
> >  However, I still have some concerns. As shown in line 53 of the revision paper, one motivation for designing the OCBC method is that the TD-based method is highly sensitive to hyperparameters. However, as shown in the reply, the GCReinSL seems also to be a parameter-sensitive method, which can not fully support the discussion in the introduction. It would be better to discuss how to choose the hyperparameters of the GCReinSL. I will maintain my score.

---

### Official Review · Reviewer_TUes · 2024-11-03

**Soundness:** 2
**Presentation:** 1
**Contribution:** 1
**Rating:** 1
**Confidence:** 4

**Summary:**

The paper aims to enhance the effectiveness of supervised learning (SL) methods in reinforcement learning (RL) by introducing a framework called Goal-Conditioned Reinforced Supervised Learning (GCReinSL). The authors propose that traditional SL-based RL methods, such as outcome-conditioned behavioral cloning (OCBC), lack trajectory stitching capabilities, which are critical for integrating suboptimal trajectories into optimal paths—a feature common in temporal-difference (TD) learning. To address this, the authors introduce Q-conditioned maximization, positing that the objective in goal-conditioned RL is equivalent to the Q-function, thereby allowing SL methods to maximize expected returns.

The paper presents GCReinSL as a solution to bridge the gap between SL and TD learning by embedding Q-function maximization into SL-based methods. The proposed approach is evaluated on various goal-conditioned offline RL tasks, such as Pointmaze and Antmaze, and compared against other methods like IQL, CQL, and other sequence modeling techniques. The authors claim that GCReinSL improves stitching performance and generalization across unseen goal-state pairs in offline RL datasets.

**Strengths:**

- The paper tackles the relevant challenge of bridging the gap between supervised learning (SL) and temporal-difference (TD) learning, especially focusing on trajectory stitching—a key limitation of SL-based RL methods.
- The paper’s focus on goal-conditioned RL is timely and aligns with practical applications in areas like robotics and offline RL.

**Weaknesses:**

- **Inconsistent and Incomplete Notations**:
The mathematical notations are poorly defined and inconsistent, for example, equations like Eq. (3), which omits necessary terms such as the expectation over the initial state distribution. These issues create significant barriers to understanding the proposed approach.

- **Lack of Theoretical Rigor**:
Theorem 4.1 and its proof are presented in a sloppy and non-rigorous manner. Important terms are either undefined or unclear.

- **Underwhelming Empirical Performance**:
The proposed method, GCReinSL, underperforms significantly compared to existing methods like IQL and CQL, particularly in the more challenging Antmaze datasets. The results fail to justify the claimed advantages of sequence modeling approaches over TD-based methods.

- I have spent a significant amount of time and effort thoroughly reviewing this paper, but the conceptual, theoretical, and empirical weaknesses, along with poor clarity, lead me to come to a conclusion that the paper is not ready for publication.

- **Lack of Clarity**:
The paper is riddled with errors and unclear explanations, making it challenging to read (see below). The poor writing quality detracts from the overall presentation and makes it difficult to follow the core ideas.


### **Comments & questions**

- Line 56-57: How come “the objective in goal-conditioned RL is equivalent to the Q-function”? How does a function “Q-function” is equivalent to an objective (either learning or optimization)? Perhaps, the authors intend to say “the objective function?” Objectives and objective functions are two different things.
- Line 67” What is “in-distribution Q-function?” Please properly define what you mean by “in-distribution” (perhaps with respect to offline data set?)
- Line 69: Is it “predicted Q-function” or “estimated Q-function”?
- Line 70: What is “the current maximum Q-function?” What do you mean by current? By “maximum Q-function” do you mean max of Q-functions or max_a Q(s,a)? The latter is NOT maximum Q-function… is maximum Q-value.
- In Eq.(3), expectation with respect to initial state distribution is NOT included in $J(\pi)$? Then, shouldn’t $J(\pi)$ be also function of s? Also, in Eq.(2), when you define conditional “discounted state occupancy distribution,” $\pi$ is used as a goal-conditioned policy $\pi(a, | s, g)$ is conditioned on a pair of state and goal. Now, in (3), the authors are using trajectory-wise policy in (3), particularly the conditional distribution within the expectation of (3), which is not consistent with the definition of (2) The authors should avoid any unnecessary overloading, and be clear about what $\pi$ they use. Perhaps, the authors need to appropriately define the relationship  between $\pi(\tau | g)$ and $\pi(a, | s, g)$.
- In any real-world environment, how does an agent observe rewards as “the (exact) probability of reaching the goal at the next time stee” as defined in Eq.(4)? Do the trajectories defined in 171-172 contain these probability rewards as offline data?
- The conditional distribution “p^\pi_+(s_{t+} = g | s_0 = s,a)$ in Eq.(6) conditioned on state and action pair has been properly defined.
- Line 216-217, what is $\hat{Q}_t$? Is it $\hat{Q}_t = \hat{Q}(s_t, a_t)$? Please be precise.
- Line 216-219, the explanation is not clear. For example, the authors explain that OCBC methods will reach the state $g_1$ rather then $g$ since since Q-function is still zero. But, there are no explonations as to why… They say that “$\hat{Q}_t = 1$ is impossible to obtain given $\hat{Q}_0 = 0$. Please clearly explain why.
- Line 235: When introducing the “Latent Variable Model,” please define what $\psi$ is.
- What is $p(z|s)$ in Eq.(8)? The authors only mentioned $p(z|s,a) = \mathcal{N}(0,I)$ as a prior.
- Line 246-247: In “we can approximate the probability $p^{\pi(\cdot | \cdot ,g)}(s_{t+} = g | s, a)$ in Eq.(6) by $−\mathcal{L}_\text{ELBO}$”, how?
- What is $t$ in the expectation $E_t$ in Eq.(10)? There is NO mention of $t$ in the entire section of 4.3.
- In Theorem 4.1, after defining $\textbf{SG}= (s, g, a, Q)$, the authors write $Q(\textbf{SG}, a)$ for $Q_\text{max}$. So, $Q(\textbf{SG}, a) = Q(s, g, a, Q, a) $? At this point (along with the previous unclear presentation), the paper becomes very difficult to read…
- In Theorem 4.1, so $\mathbf{Q}^m$ is a policy?
- In Theorem 4.1, $\pi_{\theta}^* = \arg \min \mathcal{L}^m_Q$ what is the $\arg \min$ over? All of sudden, the authors introduce $\theta$ which I suppose is the parameter for the policy, then they need to define it. Also, if $\pi_{\theta}^*$ is function of $\theta$, but the loss $\mathcal{L}^m_Q$ does not depend on the parameter of a policy? The authors are very sloppy about the notations and presentations throughout the paper, yet even one of their main results (Theorem 4.1) fails to provide meaningful contributions particularly with its non-rigorous presentation.
- In the proof of Theorem 4.1 in Appendix A.2, aren’t $\mathbf{Q}^m$ a vector (or even matrix)? How do you define inequality between vectors, is it element-wise? Then, the authors should be explicit about that. What do you mean by “all Q-values from the offline dataset” in 917? All true Q-values? The proof of Theorem 4.1 makes difficult to validate its claim.
- After reading the proof of Theorem 4.2 in Appendix A.3 and also considering the statement itself, I do not think Theorem 4.2. is a proper mathematical theorem. “Remark” (or corollary at best) would be an adequate category.

### **Minor errors**
Line 23: “by incorporating maximize” => “by incorporating maximizing?”
Line 53 Acronym “DT” is used without properly defining what it is first.
Line 54-55: The citation “Zhuang et al. (2024)” should be used with \citep
Line 75: Acronym “RvS is used without properly defining what it is first.
Line 145: “maximise” or “maximize” like the other expressions in the text? It would be good to maintain the style consistency.
Line 162: “Q-function are” => “Q-functions are”
Line 173: Perhaps,  “In each $\tau_i$ for $i in 1, …, N$“ would be more clear.
Line 177: “provide present” => “present”
Line 220: “OOD” => “out-of-distribution (OOD)” please define acronyms when first introduced.
Line 262: “After estimate the Q-function” => “After estimating the Q-function”
Line 274-275: “more weights to the $Q$ larger than” is missing $\hat{Q}$.

**Questions:**

See the questions above.

---

> ### Author Response · Authors · 2024-12-02
> **Author Responses**
>
> Thank you for the detailed review and for the suggestions for improving the work.
>
> We have carefully reviewed the issues you pointed out and made the corresponding revisions (the modified sections are highlighted in light blue. If a section title is highlighted in light blue, it indicates that the entire section has been revised). Additionally, there are a few points we would like to address.
>
> - An in-distribution Q-function refers to a Q-function learned on an offline dataset that does not produce out-of-distribution (OOD) values.
>
> - In Theorem 4.1, $Q^m$ refers to the predicted Q-values output by the model. The intention is to clarify that $Q^m$ represents the direct output of the model itself. For instance, in the case of Decision Transformer (DT), it similarly produces the corresponding return values as part of its own output.
>
> - The term "OOD" mentioned in line 220 has already been defined earlier in line 193.
>
> - I have addressed all the remaining issues based on your suggestions. If there are any points that are still unclear or require further clarification, I welcome further discussion.

---

### Official Review · Reviewer_hWfc · 2024-11-06

**Soundness:** 3
**Presentation:** 3
**Contribution:** 3
**Rating:** 6
**Confidence:** 3

**Summary:**

This paper proposes a novel method named Goal-Conditioned Reinforced Supervised Learning (GCReinSL) aiming to address the limitation of outcome-conditioned behavioral cloning (OCBC) methods in reinforcement learning tasks. Current supervised learning methods in RL lack the capability of trajectory stitching which allows the algorithms to effectively combine data from suboptimal trajectories to achieve better performance. This paper leverages expectile regression for Q-function estimation and demonstrates through theoretical analysis and experiments that this augmentation enables OCBC methods to solve the stitching problem. Experimental results on offline datasets show that GCReinSL outperforms existing goal-conditioned SL methods.

**Strengths:**

- The paper solves the stitching problem in OCBC methods by introducing Q-conditioned maximization, which allows the algorithm to combine data from suboptimal trajectories.
- The paper provides theoretical and empirical analysis to demonstrate the effectiveness of the proposed method in enhancing OCBC methods.
- The motivation for using executive regression for Q-function estimation is well-explained and aligns with the goal of estimating the maximum expected return without out-of-distribution issues.

**Weaknesses:**

- The method needs to learn a conditional variational autoencoder which could introduce additional computational overhead and complexity.
- The proposed method is constrained by the goal-conditioned formulation which may limit its application.

**Questions:**

- Can you provide more insights into the computational cost of adding the conditional variational autoencoder and how it scales with the size of the dataset?
- Can this method adapt to the return-conditioned formulation? What are the challenges and limitations of this approach?

---

### Meta-Review · Area_Chair_4faq · 2024-12-21

**Metareview:**

This paper studies reinforcement learning via supervised learning and explores how to equip supervised learning with trajectory stitching ability. The authors propose Goal-Conditioned Reinforced Supervised Learning (GCReinSL), which emphasizes maximizing the Q-function during training to estimate the maximum expected return within the distribution, subsequently guiding optimal action selection during inference. However, based on the reviewers’ feedback and the discussion with the authors, several concerns remain unresolved. Firstly, the title appears to be overclaiming. More importantly, there is a lack of mathematical rigor in the proofs, with inconsistent notations and numerous errors undermining the theoretical analysis’s credibility. Secondly, the experimental results reveal a significant performance gap compared to TD-based algorithms, contradicting the authors’ claim of significantly narrowing this gap. The authors did not provide effective responses to address these critical issues during the rebuttal phase. Therefore, I recommend rejection of this paper and encourage the authors to significantly revise their work for future submissions.

**Additional Comments On Reviewer Discussion:**

The reviewers noted that the title of the paper is overclaiming. Furthermore, there is a lack of mathematical rigor in the proofs, with inconsistent notations and numerous errors, which undermines the credibility of the theoretical analysis. Additionally, the experimental results reveal a significant gap compared to TD-based algorithms, contradicting the authors’ claim of significantly narrowing this gap. The authors did not provide detailed or satisfactory responses to address these issues during the rebuttal phase.

---

### Decision · Program_Chairs · 2025-01-22

Reject